# Flexpoint: An Adaptive Numerical Format for Efficient Training of Deep Neural Networks

**Urs Köster**[*][†], **Tristan J. Webb,**[*] **Xin Wang,**[*] **Marcel Nassar,**[*] **Arjun K. Bansal, William H. Constable, Oğuz H. Elibol, Scott Gray,**[‡] **Stewart Hall,**[†] **Luke Hornof, Amir Khosrowshahi, Carey Kloss, Ruby J. Pai, Naveen Rao**

Artificial Intelligence Products Group, Intel Corporation

## Abstract

Deep neural networks are commonly developed and trained in 32-bit floating point format. Significant gains in performance and energy efficiency could be realized by training and inference in numerical formats optimized for deep learning. Despite advances in limited precision inference in recent years, training of neural networks in low bit-width remains a challenging problem. Here we present the Flexpoint data format, aiming at a complete replacement of 32-bit floating point format training and inference, designed to support modern deep network topologies without modifications. Flexpoint tensors have a shared exponent that is dynamically adjusted to minimize overflows and maximize available dynamic range. We validate Flexpoint by training AlexNet [1], a deep residual network [2, 3] and a generative adversarial network [4], using a simulator implemented with the *neon* deep learning framework. We demonstrate that 16-bit Flexpoint closely matches 32-bit floating point in training all three models, without any need for tuning of model hyperparameters. Our results suggest Flexpoint as a promising numerical format for future hardware for training and inference.

## 1 Introduction

Deep learning is a rapidly growing field that achieves state-of-the-art performance in solving many key data-driven problems in a wide range of industries. With major chip makers' quest for novel hardware architectures for deep learning, the next few years will see the advent of new computing devices optimized for training and inference of deep neural networks with increasing performance at decreasing cost.

Typically deep learning research is done on CPU and/or GPU architectures that offer native 64-bit, 32-bit or 16-bit floating point data format and operations. Substantial improvements in hardware footprint, power consumption, speed, and memory requirements could be obtained with more efficient data formats. This calls for innovations in numerical representations and operations specifically tailored for deep learning needs.

Recently, inference with low bit-width fixed point data formats has made significant advancement, whereas low bit-width training remains an open challenge [5, 6, 7]. Because training in low precision reduces memory footprint and increases the computational density of the deployed hardware infrastructure, it is crucial to efficient and scalable deep learning applications.

---

[*]Equal contribution.
[†]Currently with Cerebras Systems, work done while at Nervana Systems and Intel Corporation.
[‡]Currently with OpenAI, work done while at Nervana Systems.

In this paper, we present Flexpoint, a flexible low bit-width numerical format, which faithfully maintains algorithmic parity with full-precision floating point training and supports a wide range of deep network topologies, while at the same time substantially reduces consumption of computational resources, making it amenable for specialized training hardware optimized for field deployment of already existing deep learning models.

The remainder of this paper is structured as follows. In Section 2, we review relevant work in literature. In Section 3, we present the Flexpoint numerical format along with an exponent management algorithm that tracks the statistics of tensor extrema and adjusts tensor scales on a per-minibatch basis. In Section 4, we show results from training several deep neural networks in Flexpoint, showing close parity to floating point performance: AlexNet and a deep residual network (ResNet) for image classification, and the recently published Wasserstein GAN. In Section 5, we discuss specific advantages and limitations of Flexpoint, and compare its merits to those of competing low-precision training schemes.

## 2 Related Work

In 2011, Vanhoucke *et al.* first showed that inference and training of deep neural networks is feasible with values of certain tensors quantized to a low-precision fixed point format [8]. More recently, an increasing number of studies demonstrated low-precision inference with substantially reduced computation. These studies involve, usually in a model-dependent manner, quantization of specific tensors into low-precision fixed point formats. These include quantization of weights and/or activations to 8-bit [8, 9, 10, 11], down to 4-bit, 2-bit [12, 13] or ternary [10], and ultimately all binary [7, 14, 5, 6]. Weights trained at full precision are commonly converted from floating point values, and bit-widths of component tensors are either pre-determined based on the characteristics of the model, or optimized per layer [11]. Low-precision inference has already made its way into production hardware such as Google's tensor processing unit (TPU) [15].

On the other hand, reasonable successes in low-precision training have been obtained with binarized [13, 16, 17, 5] or ternarized weights [18], or binarized gradients in the case of stochastic gradient descent [19], while accumulation of activations and gradients is usually at higher precision. Motivated by the non-uniform distribution of weights and activations, Miyashita *et al.* [20] used a logarithmic quantizer to quantize the parameters and gradients to 6 bits without significant loss in performance. XNOR-nets focused on speeding up neural network computations by parametrizing the activations and weights as rank-1 products of binary tensors and higher precision scalar values [7]. This enables the use of kernels composed of XNOR and bit-count operations to perform highly efficient convolutions. However, additional high-precision multipliers are still needed to perform the scaling after each convolution which limits its performance. Quantized Neural Networks (QNNs), and their binary version (Binarized Nets), successfully perform low-precision inference (down to 1-bit) by keeping real-valued weights and quantizing them only to compute the gradients and performing forward inference [17, 5]. Hubara *et al.* found that low precision networks coupled with efficient bit shift-based operations resulted in computational speed-up, from experiments performed using specialized GPU kernels. DoReFa-Nets utilize similar ideas as QNNs and quantize the gradients to 6-bits to achieve similar performance [6]. The authors also trained in limited precision the deepest ResNet (18 layers) so far.

The closest work related to this manuscript is by Courbariaux *et al.* [21], who used a dynamical fixed point (DFXP) format in training a number of benchmark models. In their study, tensors are polled periodically for the fraction of overflowed entries in a given tensor: if that number exceeds a certain threshold the exponent is incremented to extend the dynamic range, and vice versa. The main drawback is that this update mechanism only passively reacts to overflows rather than anticipating and preemptively avoiding overflows; this turns out to be catastrophic for maintaining convergence of the training.

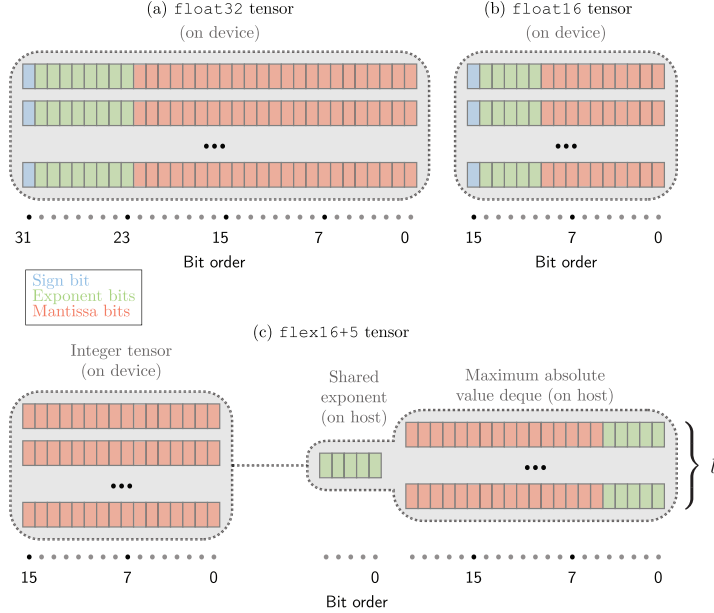

Figure 1: Diagrams of bit representations of different tensorial numerical formats. Red, green and blue shading each signify mantissa, exponent, and sign bits respectively. In both (a) **IEEE 754** 32-bit floating point and (b) **IEEE 754** 16-bit floating point a portion of the bit string are allocated to specify exponents. (c) illustrates a Flexpoint tensor with 16-bit mantissa and 5-bit shared exponent.

# 3 Flexpoint

## 3.1 The Flexpoint Data Format

Flexpoint is a data format that combines the advantages of fixed point and floating point arithmetic. By using a common exponent for integer values in a tensor, Flexpoint reduces computational and memory requirements while automatically managing the exponent of each tensor in a user transparent manner.

Flexpoint is based on tensors with an $N$-bit mantissa storing an integer value in two's complement form, and an $M$-bit exponent $e$, shared across all elements of a tensor. This format is denoted as `flexN+M`. Fig. 1 shows an illustration of a Flexpoint tensor with a 16-bit mantissa and 5-bit exponent, i.e. `flex16+5` compared to 32-bit and 16-bit floating point tensors. In contrast to floating point, the exponent is shared across tensor elements, and different from fixed point, the exponent is updated automatically every time a tensor is written.

Compared to 32-bit floating point, Flexpoint reduces both memory and bandwidth requirements in hardware, as storage and communication of the exponent can be amortized over the entire tensor. Power and area requirements are also reduced due to simpler multipliers compared to floating point. Specifically, multiplication of entries of two separate tensors can be computed as a fixed point operation since the common exponent is identical across all the output elements. For the same reason, addition across elements of the same tensor can also be implemented as fixed point operations. This essentially turns the majority of computations of deep neural networks into fixed point operations.

## 3.2 Exponent Management

These remarkable advantages come at the cost of added complexity of exponent management and dynamic range limitations imposed by sharing a single exponent. Other authors have reported on the range of values contained within tensors during neural network training: "the activations, gradients and parameters have very different ranges" and "gradients ranges slowly diminish during the training" [21]. These observations are promising indicators on the viability of numerical formats based around tensor shared exponents. Fig. 2 shows histograms of values from different types of tensors taken from a 110-layer ResNet trained on CIFAR-10 using 32-bit floating point.

In order to preserve a faithful representation of floating point, tensors with a shared exponent must have a sufficiently narrow dynamic range such that mantissa bits alone can encode variability. As suggested by Fig. 2, 16-bits of mantissa is sufficient to cover the majority of values of a single tensor. For performing operations such as adding gradient updates to weights, there must be sufficient mantissa overlap between tensors, putting additional requirements on number of bits needed to represent values in training, as compared to inference. Establishing that deep learning tensors conform to these requirements during training is a key finding in our present results. An alternative solution to addressing this problem is stochastic rounding [22].

Finally, to implement Flexpoint efficiently in hardware, the output exponent has to be determined before the operation is actually performed. Otherwise the intermediate result needs to be stored in high precision, before reading the new exponent and quantizing the result, which would negate much of the potential savings in hardware. Therefore, intelligent management of the exponents is required.

### 3.3 Exponent Management Algorithm

We propose an exponent management algorithm called Autoflex, designed for iterative optimizations, such as stochastic gradient descent, where tensor operations, e.g. matrix multiplication, are performed repeatedly and outputs are stored in hardware buffers. Autoflex predicts an optimal exponent for the output of each tensor operation based on tensor-wide statistics gathered from values computed in previous iterations.

The success of training in deep neural networks in Flexpoint hinges on the assumption that ranges of values in the network change sufficiently slowly, such that exponents can be predicted with high accuracy based on historical trends. If the input data is independently and identically distributed, tensors in the network, such as weights, activations and deltas, will have slowly changing exponents. Fig. 3 shows an example of training a deep neural network model.

The Autoflex algorithm tracks the maximum absolute value $\Gamma$, of the mantissa of every tensor, by using a dequeue to store a bounded history of these values. Intuitively, it is then possible to estimate a trend in the stored values based on a statistical model, use it to anticipate an overflow, and increase the exponent preemptively to prevent overflow. Similarly, if the trend of $\Gamma$ values decreases, the exponent can be decreased to better utilize the available range.

We formalize our terminology as follows. After each kernel call, statistics are stored in the floating point representation $\phi$ of the maximum absolute values of a tensor, obtained as $\phi = \Gamma\varkappa$, by multiplying the maximum absolute mantissa value $\Gamma$ with scale factor $\varkappa$. This scale factor is related to the exponent $e$ by the relation $\varkappa = 2^{-e}$.

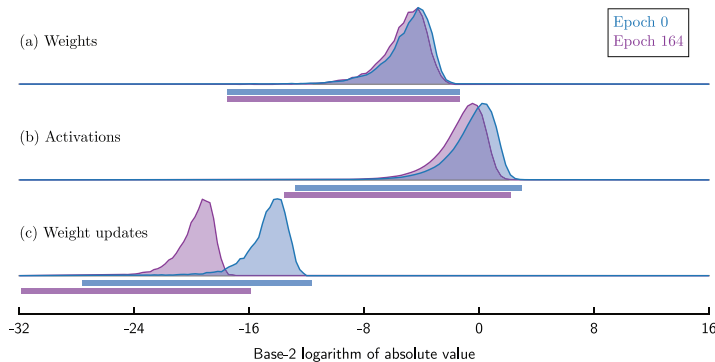

Figure 2: Distributions of values for (a) weights, (b) activations and (c) weight updates, all during the first epoch (blue) and last epoch (purple) of training a ResNet trained on CIFAR-10 for 165 epochs. The horizontal axis covers the entire range of values that can be represented in 16-bit Flexpoint, with the horizontal bars indicating the dynamic range covered by the 16-bit mantissa. All tensors have a narrow peak close to the right edge of the horizontal bar, where values have close to the same precision as if the elements had individual exponents.

If the same tensor is reused for different computations in the network, we track the exponent $e$ and the statistics of $\phi$ separately for each use. This allows the underlying memory for the mantissa to be shared across different uses, without disrupting the exponent management.

## 3.4 Autoflex Initialization

At the beginning of training, the statistics queue is empty, so we use a simple trial-and-error scheme described in Algorithm 1 to initialize the exponents. We perform each operation in a loop, inspecting the output value of $\Gamma$ for overflows or underutilization, and repeat until the target exponent is found.

---

**Algorithm 1** Autoflex initialization algorithm. Scales are initialized by repeatedly performing the operation and adjusting the exponent up in case of overflows or down if not all bits are utilized.

---

1: initialized $\leftarrow$ False
2: $\varkappa = 1$
3: **procedure** INITIALIZE SCALE
4:     **while** not initialized **do**
5:         $\Gamma \leftarrow$ returned by kernel call
6:         **if** $\Gamma \geq 2^{N-1} - 1$ **then**                                     $\triangleright$ overflow: increase scale $\varkappa$
7:            $\varkappa \leftarrow \varkappa \times 2^{\lfloor \frac{N-1}{2} \rfloor}$
8:         **else if** $\Gamma < 2^{N-2}$ **then**                               $\triangleright$ underflow: decrease scale $\varkappa$
9:            $\varkappa \leftarrow \varkappa \times 2^{\lceil \log_2 \max{(\Gamma, 1)} \rceil - (N-2)}$       $\triangleright$ Jump directly to target exponent
10:            **if** $\Gamma > 2^{\lfloor \frac{N-1}{2} \rfloor - 2}$ **then**         $\triangleright$ Ensure enough bits for reliable jump
11:                initialized $\leftarrow$ True
12:         **else**                                          $\triangleright$ scale $\varkappa$ is correct
13:            initialized $\leftarrow$ True

---

## 3.5 Autoflex Exponent Prediction

After the network has been initialized by running the initialization procedure for each computation in the network, we train the network in conjunction with a scale update Algorithm 2 executed twice per minibatch, once after forward activation and once after backpropagation, for each tensor / computation in the network. We maintain a fixed length dequeue **f** of the maximum floating point values encountered in the previous $l$ iterations, and predict the expected maximum value for the next iteration based on the maximum and standard deviation of values stored in the dequeue. If an overflow is encountered, the history of statistics is reset and the exponent is increased by one additional bit.

---

**Algorithm 2** Autoflex scaling algorithm. Hyperparameters are multiplicative headroom factor $\alpha = 2$, number of standard deviations $\beta = 3$, and additive constant $\gamma = 100$. Statistics are computed over a moving window of length $l = 16$. Returns expected maximum $\varkappa$ for the next iteration.

---

1: $\mathbf{f} \leftarrow$ stats dequeue of length $l$
2: $\Gamma \leftarrow$ Maximum absolute value of mantissa, returned by kernel call
3: $\varkappa \leftarrow$ previous scale value $\varkappa$
4: **procedure** ADJUST SCALE
5:     **if** $\Gamma \geq 2^{N-1} - 1$ **then**                    $\triangleright$ overflow: add one bit and clear stats
6:         clear **f**
7:         $\Gamma \leftarrow 2\Gamma$
8:     $\mathbf{f} \leftarrow [\mathbf{f}, \Gamma \varkappa]$                               $\triangleright$ Extend dequeue
9:     $\chi \leftarrow \alpha \left[ \max(\mathbf{f}) + \beta \mathrm{std}(\mathbf{f}) + \gamma \varkappa \right]$      $\triangleright$ Predicted maximum value for next iteration
10:     $\varkappa \leftarrow 2^{\lceil \log_2 \chi \rceil - N + 1}$                        $\triangleright$ Nearest power of two

---

## 3.6 Autoflex Example

We illustrate the algorithm by training a small 2-layer perceptron for 400 iterations on the CIFAR-10 dataset. During training, $\varkappa$ and $\Gamma$ values are stored at each iteration, as shown in Fig. 3, for instance, a linear layer's weight, activation, and update tensors. Fig. 3(a) shows the weight tensor, which is highly stable as it is only updated with small gradient steps. $\Gamma$ slowly approaches its maximum

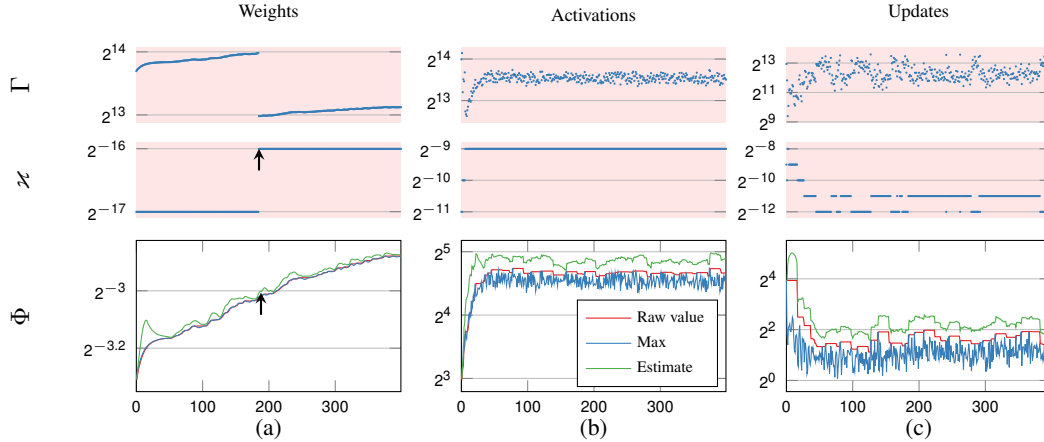

Figure 3: Evolution of different tensors during training with corresponding mantissa and exponent values. The second row shows the scale $\varkappa$, adjusted to keep the maximum absolute mantissa values ($\Gamma$, first row) at the top of the dynamic range without overflowing. As the product of the two ($\Phi$, third row) is anticipated to cross a power of two boundary, the scale is changed so as to keep the mantissa in the correct range. (a) Shows this process for a weight tensor, which is very stable and slowly changing. The black arrow indicates how scale changes are synchronized with crossings of the exponent boundary. (b) shows an activation tensor with a noisier sequence of values. (c) shows a tensor of updates, which typically displays the most frequent exponent changes. In each case the Autoflex estimate (green line) crosses the exponent boundary (gray horizontal line) before the actual data (red) does, which means that exponent changes are predicted before an overflow occurs.

value of $2^{14}$, at which point the $\varkappa$ value is updated, and $\Gamma$ drops by one bit. Shown below is the corresponding floating point representation of the statistics computed from $\Phi$, which is used to perform the exponent prediction. Using a sliding window of 16 values, the predicted maximum is computed, and used to set the exponent for the next iteration. In Fig. 3(a), the prediction crosses the exponent boundary of $2^3$ about 20 iterations before the value itself does, safely preventing an overflow. Tensors with more variation across epochs are shown in Fig. 3(b) (activations) and Fig. 3(c) (updates). The standard deviation across iterations is higher, therefore the algorithm leaves about half a bit and one bit respectively of headroom. Even as the tensor fluctuates in magnitude by more than a factor of two, the maximum absolute value of the mantissa $\Gamma$ is safely prevented from overflowing. The cost of this approach is that in the last example $\Gamma$ reaches 3 bits below the cutoff, leaving the top bits zero and using only 13 of the 16 bits for representing data.

### 3.7 Simulation on GPU

The experiments described below were performed on Nvidia GPUs using the *neon* deep learning framework[4]. In order to simulate the `flex16+5` data format we stored tensors using an `int16` type. Computations such as convolution and matrix multiplication were performed with a set of GPU kernels which convert the underlying `int16` data format to `float32` by multiplying with $\varkappa$, perform operations in floating point, and convert back to `int16` before returning the result as well as $\Gamma$. The kernels also have the ability to compute only $\Gamma$ without writing any outputs, to prevent writing invalid data during exponent initialization. The computational performance of the GPU kernels is comparable to pure floating point kernels, so training models in this Flexpoint simulator adds little overhead.

## 4 Experimental Results

### 4.1 Convolutional Networks

We trained two convolutional networks in `flex16+5`, using `float32` as a benchmark: AlexNet [1], and a ResNet [2, 3]. The ResNet architecture is composed of modules with shortcuts in the dataflow

graph, a key feature that makes effective end-to-end training of extremely deep networks possible. These multiple divergent and convergent flows of tensor values at potentially disparate scales might pose unique challenges for training in fixed point numerical format.

We built a ResNet following the design as described in [3]. The network has 12 blocks of residual modules consisting of convolutional stacks, making a deep network of 110 layers in total. We trained this model on the CIFAR-10 dataset [1] with `float32` and `flex16+5` data formats for 165 epochs.

Fig. 4 shows misclassification error on the validation set plotted over the course of training. Learning curves match closely between `float32` and `flex16+5` for both networks. In contrast, models trained in `float16` without any changes in hyperparameter values substantially underperformed those trained in `float32` and `flex16+5`.

## 4.2 Generative Adversarial Networks

Next, we validate training a generative adversarial network (GAN) in `flex16+5`. By virtue of an adversarial (two-player game) training process, GAN models provide a principled way of unsupervised learning using deep neural networks. The unique characteristics of GAN training, namely separate data flows through two components (generator and discriminator) of the network, in addition to feeds of alternating batches of real and generated data of drastically different statistics to the discriminator at early stages of the training, pose significant challenges to fixed point numerical representations.

We built a Wasserstein GAN (WGAN) model [4], which has the advantage of a metric, namely the Wasserstein-1 distance, that is indicative of generator performance and can be estimated from discriminator output during training. We trained a WGAN model with the LSUN [23] bedroom dataset in `float32`, `flex16+5` and `float16` formats with exactly the same hyperparameter settings. As shown in Fig. 5(a), estimates of the Wasserstein distance in `flex16+5` training and in `float32` training closely tracked each other. In `float16` training the distance deviated significantly from baseline `float32`, starting with an initially undertrained discriminator. Further, we found no differences in the quality of generated images between `float32` and `flex16+5` at specific stages of the training 5(b), as quantified by the Fréchet Inception Distance (FID) [24]. Generated images from `float16` training had lower quality (significantly higher FIDs, Fig. 5(b)) with noticeably more saturated patches, examples illustrated in Fig. 5(c), 5(d) and 5(e).

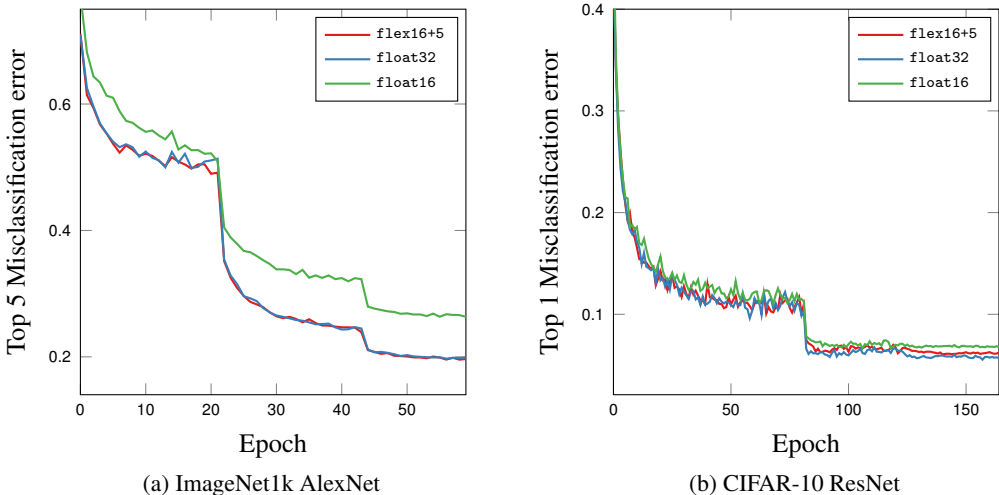

(a) ImageNet1k AlexNet        (b) CIFAR-10 ResNet

Figure 4: Convolutional networks trained in `flex16+5` and `float32` numerical formats. (a) AlexNet trained on ImageNet1k, graph showing top-5 misclassification on the validation set. (b) ResNet of 110 layers trained on CIFAR-10, graph showing top-1 misclassification on the validation set.

# 5   Discussion

In the present work, we show that a Flexpoint data format, `flex16+5`, can adequately support training of modern deep learning models without any modifications of model topology or hyperparameters, achieving a numerical performance on par with `float32`, the conventional data format widely used in deep learning research and development. Our discovery suggests a potential gain in efficiency and performance of future hardware architectures specialized in deep neural network training.

Alternatives, i.e. schemes that more aggressively quantize tensor values to lower bit precisions, also made significant progress recently. Here we list major advantages and limitations of Flexpoint, and make a detailed comparison with competing methods in the following sections.

Distinct from very low precision (below 8-bit) fixed point quantization schemes which significantly alter the quantitative behavior of the original model and thus requires completely different training algorithms, Flexpoint's philosophy is to maintain numerical parity with the original network training behavior in high-precision floating point. This brings about a number of advantages. First, all prior knowledge of network design and hyperparameter tuning for efficient training can still be fully leveraged. Second, networks trained in high-precision floating point formats can be readily deployed in Flexpoint hardware for inference, or as component of a bigger network for training. Third, no re-tuning of hyperparameters is necessary for training in Flexpoint–what works with floating point simply works in Flexpoint. Fourth, the training procedure remains exactly the same, eliminating the need of intermediate high-precision representations, with the only exception of intermediate higher

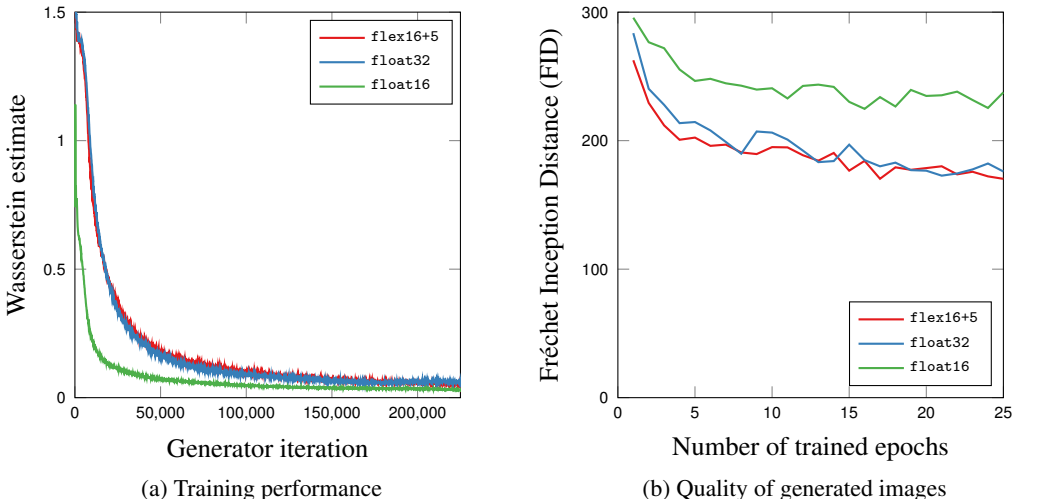

(a) Training performance                    (b) Quality of generated images

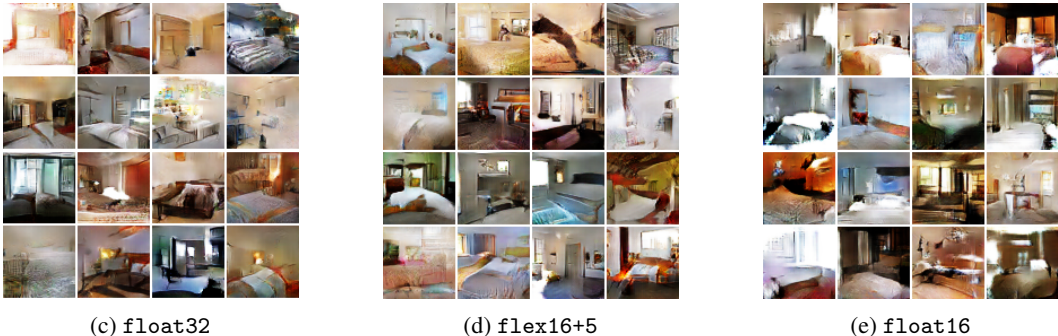

(c) `float32`                    (d) `flex16+5`                    (e) `float16`

Figure 5: Training performance of WGAN in `flex16+5`, `float32` and `float16` data formats. (a) Learning curves, i.e. estimated Wasserstein distance by median filtered and down-sampled values of the negative discriminator cost function, median filter kernel length 100 [4], and down-sampling by plotting every 100th value. Examples of generated images by the WGAN trained with in (c) `float32`, (d) `flex16+5` and (e) `float16` for 16 epochs. Fréchet Inception Distance (FID) estimated from 5000 samples of the generator, as in [24].

precision accumulation commonly needed for multipliers and adders. Fifth, all Flexpoint tensors are managed in exactly the same way by the Autoflex algorithm, which is designed to be hidden from the user, eliminating the need to remain cognizant of different type of tensors being quantized into different bit-widths. And finally, the AutoFlex algorithm is robust enough to accommodate diverse deep network topologies, without the need of model-specific tuning of its hyperparameters.

Despite these advantages, the same design philosophy of Flexpoint likely prescribes a potential limitation in performance and efficiency, especially when compared to more aggressive quantization schemes, e.g. Binarized Networks, Quantized Networks and the DoReFa-Net. However, we believe Flexpoint strikes a desirable balance between aggressive extraction of performance and support for a wide collection of existing models. Furthermore, potentials and implications for hardware architecture of other data formats in the Flexpoint family, namely `flexN+M` for certain $(N, M)$, are yet to be explored in future investigations.

**Low-precision data formats:** TensorFlow provides tools to quantize networks into 8-bit for inference [9]. TensorFlow's numerical format shares some common features with Flexpoint: each tensor has two variables that encode the range of the tensor's values; this is similar to Autoflex $\varkappa$ (although it uses fewer bits to encode the exponent). Then an integer value is used to represent the dynamic range with a dynamic precision.

The dynamic fixed point (DFXP) numerical format, proposed in [25], has a similar representation as Flexpoint: a tensor consists of mantissa bits and values share a common exponent. This format was used by [21] to train various neural nets in low-precision with limited success (with difficulty to match CIFAR-10 maxout nets in `float32`). DFXP diverges significantly from Flexpoint in automatic exponent management: DFXP only updates the shared exponent at intervals specified by the user (e.g. per 100 minibatches) and solely based on the number of overflows occurring. Flexpoint is more suitable for training modern networks where the dynamics of the tensors might change rapidly.

**Low-precision networks:** While allowing for very efficient forward inference, the low-precision networks discussed in Section 2 share the following shortcomings when it comes to neural network training. These methods utilize an intermediate floating point weight representation that is also updated in floating point. This requires special hardware to perform these operations in addition to increasing the memory footprint of the models. In addition, these low-precision quantizations render the models so different from the exact same networks trained in high-precision floating point formats that there is often no parity at the algorithmic level, which requires completely distinct training algorithms to be developed and optimized for these low-precision training schemes.

## 6 Conclusion

To further scale up deep learning the future will require custom hardware that offers greater compute capability, supports ever-growing workloads, and minimizes memory and power consumption. Flexpoint is a numerical format designed to complement such specialized hardware.

We have demonstrated that Flexpoint with a 16-bit mantissa and a 5-bit shared exponent achieved numerical parity with 32-bit floating point in training several deep learning models without modifying the models or their hyperparameters, outperforming 16-bit floating point under the same conditions. Thus, specifically designed formats, like Flexpoint, along with supporting algorithms, such as Autoflex, go beyond current standards and present a promising ground for future research.

## 7 Acknowledgements

We thank Dr. Evren Tumer for his insightful comments and feedback.

## Footnotes

[4]Available at `https://github.com/NervanaSystems/neon`.

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
