[Reviews · NeurIPS 2017]

Reviewer 1



This work presents the Flexpoint format as a new type of deep learning suitable floating-point format. The idea itself is interesting. The proposed algorithms are largely heuristic (thus optimality not guaranteed,) which is less desirable but still fine. The biggest and unfortunately unacceptable problem of this work is the lacking of proper comparisons. We basically only learned that flex16+5 is as good as float32 in terms of accuracy (and visual quality) in few networks, which is simply not enough. How does the equally memory efficient float16 compare to flex16+5 in the same experiments? How much faster and lower power is flex16+5 compared to float32? If specialized hardware is required for flex16+5 to run more efficiently (which is likely the case), how much savings (in operations or wall clock time, and in wattage) can it provide compared to float32, and how much cost (area, cycles, etc.) is required to support this format and related operations (i.e. algorithm 1 and 2)? While the list can go on, the comparisons above should all be essential and required. Even estimation would be better than nothing. I would suggest the authors to re-submit to later conferences with these new comparisons added.

Reviewer 2



The paper presents a new data format (Flexpoint) that is intended to replace the standard 32-bit floating point format in training and inference in deep neural networks. The new format relies on tensors, where a 5-bit exponent is shared by the whole tensor and then each tensor value is represented with a 16-bit mantissa. The data format is evaluated and compared to standard float32 using two CNNs, i.e., AlexNet (trained on ImageNet1k) and ResNet (trained on CIFAR-10), and a generative adversarial network (trained with LSUN). The results are promising, and shows that the new dataformat haa a similar performance as float32. A lot of work have already been done on how to reduce the size of the weights in neural networks. However, most of the previous work have focused on representing float32 weights as small quantized integers, e.g., int8 or int4. Thus, proposing a new floating point format sticks out (in a positive way). Further, the Flexpoint format is also targeted for training (not only testing/inference). An interesting contribution in the paper is the algorithm for automatically adjusting the value of the exponent, depending on the dynamic data range. One drawback of the paper is that it does not compare FlexPoint to, e.g., some quantized approach (or similar), that also has the goal of reducing data size of weights. Further, one strong argument for reducing the weight data size is to reduce the energy consumption (smaller / more narrow format => less energy). However, this aspect is not discussed or evaluated at all in the paper. Another, severe, drawback is related to implementation. Already in the abstract, the authors claim that Flexpoint is a promising format for future hardware implementations. However, the paper does not discuss at all how the hardware should be designed to support the new format. This aspect must be addressed before publication.

Reviewer 3



Paper provides an improvement to existing approaches for training with lower precision than single precision floating point (fp32) while sticking to the same training regimes. Novelty in how predicting the exponent results via AutoFlex in better convergence. Pros: - Good results - Promising direction for future hardware and software efforts - Using AutoFlex (predictions) to manage the exponent is the primary novelty Cons: - Missing comparison to float16. Nvidia claims they are able to train models with fp16 and some additional loss scaling. While the AutoFlex approach seems simpler, it seems useful to have this comparison. - Doesn't talk about how AutoFlex hyper parameters are picked and whether the ones listed work for all models they have tried.

Reviewer 4



I find the proposed technique interesting, but I believe that these problem might be better situated for a hardware venue. This seems to be more a contribution on hardware representations, as compared to previous techniques like xor-net which don't ask for a new hardware representation and had in mind an application to compression of networks for phones, and evaluated degradation of final performance. However, I do think that the paper could be of interest to the NIPS community as similar papers have been published in AI conferences. The paper is well written and easy to understand. The upsides of the method and the potential problems are well described and easy to follow. Improvements over previous techniques seem somewhat incremental, and for many applications it's not clear that training with a low precision format is better than training with a high precision format and then cropping down, and no experiments are performed to explore that. My opinion on the evaluation is mixed. For one, no experiments are done to compare to previous (crop to low precision after training) techniques. Second, plotting validation error seems natural, but I would also like to see test numbers. I notice that the validation result for Alexnet on ILSVRC is about 20% error, while alexnet originally reported 17% error on ILSVRC-11 and 16.4% on ILSVR-2012 test. Allowing for direct comparison would improve the paper. The results for CIFAR (~5% error) seem to be near state of the art. Qualitative GAN results appear to be good. Further Comments: After discussion, I believe that the paper could be accepted (increased to a 7 from a 6). Other reviewers have argued that the authors need to provide evidence that such a system would be implementable in hardware, be faster, and have a lower memory footprint. I do not believe that those experiments are necessary. Given the large number of network architectures that people are using currently, which ones should the authors demonstrate on? If one architecture causes problems, is the paper unacceptable? I am also somewhat confused that other reviews seem to give no value to drastically reducing the memory footprint (by easy calculation each layer is represented in half as much memory). Many of the current state of the art network architectures use the maximum GPU memory during training. While it needs more work, this technique seems to provide a no-loss answer to how to increase network capacity which is very cool indeed This is also related to my original point about venue, as it is this paper sits somewhere between AI and systems. If the authors performs further "hardware-centric" experiments it seems the paper would clearly be better suited for a hardware or systems venue. If we evaluate this work as a network compression paper then the classification accuracy demonstrates success. A network trained in half as much memory is as accurate as the full memory network. It deserves consideration by the ACs for acceptance.